# Recent Increases in Influenza-Related Hospitalizations, Critical Care Resource Use, and In-Hospital Mortality: A 10-Year Population-Based Study in South Korea

**DOI:** 10.3390/jcm11164911

**Published:** 2022-08-21

**Authors:** Tae Hwa Hong, Hyung Seok Lee, Nam-Eun Kim, Kyu Jin Lee, Yong Kyun Kim, Jung Nam An, Joo-Hee Kim, Hyung Won Kim, Sunghoon Park

**Affiliations:** 1Department of Surgery, Hallym University Sacred Heart Hospital, Anyang 14068, Korea; 2Department of Nephrology, Hallym University Sacred Heart Hospital, Anyang 14068, Korea; 3Department of Public Health Sciences, Graduate School of Public Health, Seoul National University, Seoul 08826, Korea; 4Department of Pulmonary, Allergy and Critical Care Medicine, Hallym University Sacred Heart Hospital, Anyang 14068, Korea; 5Department of Infectious Disease, Hallym University Sacred Heart Hospital, Anyang 14068, Korea

**Keywords:** influenza, hospitalization, critical care, mortality

## Abstract

**Background:** Long-term trends in influenza-related hospitalizations, critical care resource use, and hospital outcomes since the 2009 H1N1 influenza pandemic season have been rarely studied for adult populations. **Materials and Methods:** Adult patients from the Korean Health Insurance Review and Assessment Service who were hospitalized with influenza over a 10-year period (2009–2019) were analyzed. The incidence rates of hospitalization, critical care resource use, and in-hospital death were calculated using mid-year population census data. **Results:** In total, 300,152 hospitalized patients with influenza were identified (men, 35.7%; admission to tertiary hospitals, 9.4%). Although the age-adjusted hospitalization rate initially decreased since the 2009 H1N1 pandemic (52.61/100,000 population in 2009/2010), it began to increase again in 2013/2014 and reached a peak of 169.86/100,000 population in 2017/2018 (*p* < 0.001). The in-hospital mortality rate showed a similar increasing trend as the hospitalization, with a peak of 1.44/100,000 population in 2017/2018 (vs. 0.35/100,000 population in 2009/2010; *p* < 0.001). The high incidence rates of both hospitalization and in-hospital mortality were mainly attributable to patients aged ≥60 years. The rate of intensive care unit admission and the use of mechanical ventilation, continuous renal replacement therapy and vasopressors have also increased from the 2013/2014 season. The incidence of heart failure was the most frequent complication investigated, with a three-fold increase in the last two seasons since 2009/2010. In multivariate analysis adjusted for covariates, among hospitalized patients, type of hospitals and 2009 H1N1 pandemic season were associated with in-hospital mortality. **Conclusions:** We confirmed that the rates of hospitalization, critical care resource use, and in-hospital mortality by influenza have increased again in recent years. Therefore, strategies are needed to reduce infections and optimize resource use with a greater focus on older people.

## 1. Introduction

Influenza, an acute infectious respiratory illness, causes substantial morbidity and mortality, most deaths occurring in adults aged ≥ 65 years. The World Health Organization (WHO) reported that influenza causes about 3~5 million cases of severe illness and 290,000–650,000 deaths annually in the world [1].

Epidemics of influenza and its disease burden vary by different regions [2,3,4,5,6]. Hence, nationwide data from countries with different healthcare systems are needed to establish global strategies and prepare for future epidemics or pandemics. Although influenza cases have decreased for several years after the 2009 H1N1 pandemic season, some Western regions are currently seeing the increasing rates of hospitalizations and hospital deaths by seasonal influenza again [7,8,9]. Hence, given the increased healthcare burden and resource consumption during a surge of infections, an investigation of long-term changes in morbidities or mortalities caused by influenza might be worthwhile in terms of estimating disease burden and guiding future policies.

In Korea, several nationwide studies on influenza, using data from the Korean National Health Insurance Service (NHIS) or the Health Insurance Review & Assessment Service (HIRA), have been conducted [10,11,12,13]. However, trends of influenza-related hospitalizations and hospital outcomes over multiple years have not been well described so far. Besides, it remains unclear how the rates of intensive care unit (ICU) admissions and other critical care resource use have changed since the 2009 H1N1 pandemic season.

Therefore, through a population-based study using HIRA data (from 2009 to 2019), we explored the patterns of changes after 2009 H1N1 pandemic season in the rates of influenza-related hospitalizations, critical care resource use, and hospital outcomes in the Korean adult population.

## 2. Materials and Methods

### 2.1. Data Collection

Data were collected from the HIRA database that contains nationwide population-based healthcare reimbursement claims in Korea. The NHIS provides healthcare coverage, using fee-for-service payment system, to about 97% of the population in Korea [14]. The government regulates these fees, and the HIRA reviews the healthcare insurance claims and assess the quality of healthcare services. The claims database holds encrypted personal information including name, age, sex, International Classification of Diseases, the 10th Revision (ICD-10) codes for primary and secondary diagnoses, dates of admission and discharge, type of hospitals, and in-hospital mortality. Records of all medical procedures and prescriptions that are covered by the NHIS are also accumulated in the HIRA database. In this study, we identified adult patients (>18 years) who were hospitalized with influenza from October 2009 to September 2019. In Korea, the influenza season begins as early as October and lasts as late as April or May in next year. However, in the present study, all hospitalized patients with influenza diagnosis were collected year-round.

This study and a waiver of consent were approved by the Hallym University Institutional Review Board (approval No. 2020-05-018).

### 2.2. Case Definition

All the diagnosis codes for influenza (J09–J11) were extracted from the claims data if the ICD-10 codes for influenza were listed for either the primary or secondary diagnosis. For the analysis, a hospitalization was defined as an admission to a hospital for 24 h or more, and hospital stays separated by <2 days were considered to be the same hospital admission; for patients who were transferred to another hospital, data were combined using resident registration numbers and discharge dates.

### 2.3. Variables for Critical Care Resource Use

ICU admissions were identified using the Korean Classification of Diseases 7th edition codes (KCD-7; AJ001, AJ003, AJ006, AJ007, AJ010, AJ011, AJ020, AJ100, AJ200, AJ300, AC612), which is a modified version of the ICD-10. Multiple ICU admissions during the same hospitalization were considered as a single ICU admission. Comorbidities such as diabetes mellitus, cardiovascular disease, cerebrovascular disease, chronic kidney disease, chronic liver disease, malignancy, rheumatic disease, and acquired immunodeficiency syndrome (AIDS) were identified with the use of ICD-10 codes (Appendix A), and also summarized using the Charlson Comorbidity Index (CCI) [15,16,17]. Procedures with mechanical ventilation (MV; the Korean NHIS procedure codes: M0850, M0857, M0858, M0860, M5830, M5850, M5857, M5858, M5860, M5890), continuous renal replacement therapy (CRRT; O7031–O7035 and O705–O7055), and extracorporeal membrane oxygenation (ECMO; O1901–O1907) were investigated. Data on the use of vasopressors (i.e., norepinephrine, epinephrine, and dopamine), steroids (i.e., methylprednisolone, prednisone, and hydrocortisone), and oseltamivir were also obtained. Regarding the steroid use, to include patients who received steroids for acute respiratory failures, only doses of ≥40 mg methylprednisolone (or its equivalent doses of other steroids) for ≥5 days were counted, based on a previous study [18].

In addition, hospitals were divided into four types according to their supporting levels, which was based on bed capacity and range of specialties as stipulated by the Korean Health Law [19]. A ‘hospital’ was defined as a healthcare facility with >30 beds. A ‘general hospital’ is one with both >100 beds and >7 specialty departments, where the inclusion of Internal Medicine, Pediatrics, Surgery, Obstetrics and Gynecology, Pathology, Anesthesiology, and Laboratory Medicine is mandatory. A ‘tertiary hospital’ is defined as one with >20 specialty departments and serves as a teaching hospital. When a healthcare facility did not fall into any of the three types (e.g., nursing hospitals, hospitals for oriental medicine, etc.), it was categorized as ‘others’.

### 2.4. Variables for Outcomes

For hospital outcomes, the incidence of complications (myocarditis, acute myocardial infarct, heart failure, cerebral infarct and hemorrhage, and liver failure) that occurred during the hospitalization was investigated using ICD-10 codes (Appendix A). In-hospital deaths, as well as ICU deaths, were also investigated for all hospitalized patients.

### 2.5. Data Analysis

Descriptive analyses were used to summarize characteristics of the hospitalized patients with influenza. Data are presented as means ± standard deviations or median (interquartile ranges, 25th–75th percentiles) for continuous data and as frequency (percentage) for categorical data. To compare the hospitalization rates between years, the age-adjusted rates of hospitalization were calculated, using the annual mid-year population census (the Korea Statistical Information Service) as the standard [20]. We investigated the longitudinal trends over 10 years (2009–2019) in the use of critical care resources, such as ICU admission, vasopressors, MV, CRRT, and ECMO, as well as in-hospital mortality rate. For these data, Cochran–Armitage trend test was used to detect yearly trends. Lengths of hospital and ICU stay were compared using the Kruskal–Wallis test because these variables were not normally distributed. Logistic regression analysis was performed to see the effects of type of hospitals and the seasons on in-hospital mortality, adjusted for age, gender, CCI, and the use of ICU resources. All statistical analyses were performed using R version 4.0.5 (R Foundation Inc.; https://www.r-project.org/, accessed on 28 February 2022). For all analyses, *p* < 0.05 was considered to indicate statistical significance.

## 3. Results

### 3.1. Study Population

In total, 300,152 patients were hospitalized with influenza in South Korea from October 2009 to September 2019 (Appendix A). The mean age of the included patients was 51.0 ± 18.7 years, and the frequency of men was 35.7%. Diabetes mellitus was the most common underlying comorbidity, followed by chronic obstructive pulmonary disease and liver disease (Appendix A). Almost half of the patients (48.7%) were treated in general hospitals; only 9.4% of patients were treated in tertiary hospitals (Table 1 and Appendix A). Except for the 2009/2010 season, the prescription rate of oseltamivir ranged from 29.4% to 55.1% during the hospitalization (Appendix A).

### 3.2. Incidence of Hospitalization

Changes in the annual hospitalization rate per 100,000 people are depicted in Figure 1. The age-adjusted incidence of hospitalizations initially decreased after the 2009 H1N1 pandemic season but increased again in the 2013/2014 season and reached its peak during the 2017/2018 season (169.86/100,000 population), which was almost three times greater than the incidence during the 2009 H1N1 pandemic (52.61/100,000 population: Table 1). This increase was found in all age groups and all regions nationwide (Appendix A). In particular, the rate of increase was higher for older patients than for younger patients (Figure 1); the proportion of women was consistently higher than that of men during the 10-year period (64.3% vs. 35.7%).

### 3.3. Critical Care Resource Use and Complications

The rate of ICU admission decreased initially after the 2009 H1N1 pandemic season, but started to increase again in 2013/2014, with a peak rate during the 2017/2018 season (3.37/100,000 vs. 1.83/100,000 population in 2009/2010, *p* < 0.001). The rates of MV, CRRT, and vasopressor use also showed a similar increasing trend as the ICU admission rate (1.24/100,000 vs. 0.63/100,000 population for MV, *p* < 0.001; 0.22/100,000 vs. 0.10/100.000 population for CRRT, *p* < 0.001; 2.27/100.000 vs. 0.35/100,000 population for vasopressors, *p* < 0.001; 2017/2018 vs. 2009/2010, respectively) (Figure 2 and Appendix A). However, a minimal change was detected in the rate of ECMO use. Regarding the complications, heart failure and cerebral infarcts were the most common (Table 2), and their incidences have increased since the 2012/2013 season, with a three-fold increase in the 2017/2018–2018/2019 seasons compared to the 2009/2010 season (*p* < 0.001 and *p* < 0.001, respectively; Appendix A).

### 3.4. Incidence Rates of ICU and In-Hospital Mortality

The ICU and in-hospital mortality rates showed a similar trend as hospitalization rates during the 10-year period. While mortalities were lowest during the 2012/2013 season (ICU mortality: 0.04/100,000; in-hospital mortality: 0.07/100,000 population), they gradually increased to peak rates in the 2017/2018 season (0.67/100,000 and 1.44/100,000 population, respectively) (Table 2 and Figure 3). In particular, the increased in-hospital mortality rate was primarily driven by patients aged ≥60 years (Figure 3 and Appendix A). Among the hospitalized patients, a multivariable analysis adjusted for age, gender, comorbidity, steroid, mechanical ventilation, CRRT, and the use of vasopressors demonstrated that type of hospitals (i.e., ‘general’, ‘tertiary’, and ‘others’, compared to ‘hospitals’) and the 2009 H1N1 pandemic season (i.e., 2009/2010–2010/2011, compared to other seasons) were significantly associated with increased in-hospital mortality (Table 3).

## 4. Discussion

This population-based study revealed several notable findings. First, since the decline of the 2009 H1N1 pandemic influenza outbreak, the incidence rates of influenza-related hospitalizations and in-hospital mortalities have increased again from the 2013/2014 season, with a more than a two-fold increase during the 2017/2018–2018/2019 seasons, compared with the 2009 H1N1 pandemic season. Notably, the incidence rates of both hospitalization and in-hospital mortality were mainly attributable to patients aged ≥60 years. Second, similar to the changes in hospitalization and in-hospital mortality, the rate of ICU admissions and the use of MV, CRRT, and vasopressors also recently increased. Finally, the incidence of heart failure and cerebral infarct increased by approximately three-fold over the 10-year period.

Influenza with epidemic or pandemic potential can be a huge burden on healthcare services worldwide. Hence, to enable efficient infection control, international surveillance and collaboration systems are important. In many countries, both clinical and laboratory data-based surveillance systems are commonly used [22]. However, to estimate the burden of influenza, detailed data on hospitalization (i.e., severe influenza) should be collected, as in the United States (U.S.) Although the national influenza surveillance system is well organized in South Korea, the data on hospitalized patients are very limited, thus hampering better understanding of the disease burden. In this aspect, our study, highlighting the increasing rates of hospitalization and ICU resource use, is relevant and can add to the current knowledge.

Although the 2009 H1N1 influenza pandemic, which began worldwide in April 2009, was associated with high mortality [23,24], data are limited on how the seasonal influenza epidemics have changed in terms of disease severity and healthcare utilization since the pandemic season. Hence, we conducted this 10-year investigation starting the 2009 pandemic season. In South Korea, cases (of the 2009 H1N1 pandemic) began to increase in late August 2009 and lasted until December 2010 [25]. Therefore, some cases of hospitalization or mortality from the 2009/2010 season may have been missed in our study. However, studies from U.S., the United Kingdom, and China (Shanghai) indicate that since the decline of the 2009 H1N1 pandemic outbreak, the rates of hospitalization have increased again, with a peak during the 2017/2018 season [7,8,9,26]. These data are consistent with our results. Although the rate of hospitalization was higher in females than in males in our study, a previous study using the HIRA data also reported similar results [10]. However, gender difference can vary among different seasons and countries [8,9,10,27].

The current study showed that the rate of in-hospital mortality declined after the 2009/2010 season but increased again in recent years. In the U.S., reports from the Centers for Disease Control and Prevention (CDC) and a surveillance network showed a similar trend, with peak mortality during the 2017/2018 season [7,9]. Although the mortality rate has not changed noticeably in the United Kingdom, the total number of deaths has increased steadily over the past decade [8]; our findings are consistent with that trend. In contrast, among the hospitalized patients, the risk of in-hospital mortality was the highest during the 2009 H1N1 pandemic, compared to other influenza seasons in multivariable analysis. One explanation for this is that different characteristics or disease severities among hospitalized patients over the study period, as well as different circulating virus strains, may have contributed to the different risks for in-hospital mortality. Notably, we found a higher prevalence of admissions to tertiary hospitals during the 2009 H1N1 pandemic season (15.6%) than during any other season. The significant association of type of hospitals with in-hospital mortality may be associated with the fact that patients with comorbidities or high disease severity were more likely to be treated in ‘general’ or ‘tertiary hospitals’ rather than ‘hospitals’, and those who indicated do-not-resuscitation or withholding life-sustaining treatments were also likely to be treated in other type of hospitals (e.g., nursing hospitals).

Regarding the ICU resource utilization, the frequency of MV, CRRT and ECMO use, as well as the rate of ICU admission, were investigated. Considering the high costs and impacts on morbidity and mortality, these variables seem relevant to answer our research questions. However, the rates of ICU admission in our cohort (0.32~3.37/100,000 population) and the United Kingdom (0.06~0.22/100,000 population) [8] seem to be lower, compared to that in the U.S. (5.7~11.6/100,000 population) or Canada (14.4%) [9,28,29]. This may be explained by the different ICU capacities or triage among hospitals or countries. Besides, the rate of MV use in our cohort (0.4~1.6% among hospitalized patients) seems lower, compared to that of the US (15.6~25.0%), Spain (8.9%), Canada (8.6%) and Germany (3.4~6.7%) [6,9,27,29,30]. This different rate could be linked to different disease severities or proportions of do-not-intubate orders among countries. Particularly, the number of ECMO cases was very low. Although we think all billing claims were processed for all ECMO cases nationwide, a possibility of underestimation of ECMO cases cannot be ruled out. However, of note, recent reports demonstrated that influenza vaccination can reduce the severity of disease in patients with influenza or even COVID-19 [31,32,33]. Hence, the high vaccination rate (>80% in those aged 65 and older) in South Korea might contribute to this lower rate of ICU admission and MV use in our study. However, the Korean national influenza surveillance system reported an increased incidence of influenza-like illness in the 2017/2018 seasons (Appendix A**).** The CDC also indicated that the 2017/2018 season had high severity, with unusually high rates of outpatient flu-like illness, hospitalizations, and complications [34]. Increased disease severity or availability of machines may be responsible for the increasing trend of ICU resource use [35].

Regarding complications investigated, a substantial increase in the incidence of heart failure and cerebral infarcts was noted during the past decade, with an almost twofold higher rate compared with respiratory failure (indicated by MV use). It is well known that pneumonia and exacerbation of underlying pulmonary or cardiac diseases are the common complications of influenza. Hence, considering the large effect of these complications on the nation’s healthcare system, we cannot overemphasize the early diagnosis and treatment of influenza. However, the incidence of liver failure in the current study was an unexpected result. Although this is likely to be associated with underlying comorbidity or disease progression, further studies are needed to clarify this association.

As with other studies using administrative database, CCI was used to investigate the frequency of underlying comorbidities in our study, and similar to previous study, COPD, diabetes mellitus, and liver disease were the most common [30,36]. However, despite a thorough review of the HIRA data, we noted that the frequency of comorbidities was lower than anticipated. Previously, the incidence of diabetes mellitus was reported to be 13.8% of Koran adults (>30 years) [37], and a study using the Hospital-based Influenza Morbidity & Mortality Surveillance also reported that the incidence of diabetes mellitus and cardiovascular diseases was 9.8% and 7.2%, respectively [38]. Hence, we cannot exclude a possibility of underestimation of the frequency of comorbidities, which may be inherent when using administrative database. However, from the Korean Statistical Information Service data [20], the prevalence of diabetes mellitus was reported to be 1.3% among the entire hospitalized population in 2018 (unpublished data). Besides, in our study, a high proportion of patients aged <60 years (64.0%) should also be taken into account.

Although we collected data on the prescription of oseltamivir (Appendix A), many patients could presumably have received oseltamivir before hospitalization, not covered by the NHIS. For these reasons, we did not include the antiviral agent in the multivariate analysis. Regarding the steroid doses, the selection criteria seem arbitrary in our study. Although a dose of 1~2 mg/kg/day methylprednisolone has been frequently used to treat patients with acute respiratory distress syndrome in clinical practice [39], we chose a dose of 40 mg methylprednisolone (or its equivalent doses of other steroids) as the minimum dose. Besides, we also intended to exclude patients receiving long-term treatment for chronic diseases.

This study has some limitations to be considered. First, as with other studies using administrative database, there is a possibility of misclassification. Particularly, although the patients with influenza ICD-10 codes in our study were those with laboratory-confirmed cases, year-to-year heterogeneity of influenza test rates may partially explain the differences in influenza-related hospitalizations. Second, it was not possible to obtain data on influenza virus types (or subtypes) and influenza severity from the billing claims data; in contrary, studies using clinical data may increase the accuracy of diagnoses and severity. Third, although vaccination has been known to decrease the rate of hospitalization and mortality [40,41], this information was not obtainable from the HIRA data because it is not claimed (i.e., non-payment) in Korea. Fourth, to evaluate ICU resource utilizations and disease burden, analysis on the duration (days) of MV, ECMO, or CRRT might have been more informative than just proportions of patients receiving these treatments. Finally, we could not provide long-term mortality data, such as 6- or 12-month mortality rates, because the data were not available for the HIRA database. However, population-based studies like ours are very rare. Additionally, one of our strengths is that we evaluated those data during the 10-year period, facilitating better understanding of long-term trends.

## 5. Conclusions

This study highlights that after the big wave of 2009 H1N1 pandemic influenza, the rates of hospitalizations and mortality by seasonal influenza have risen again in recent years, both of which were mostly driven by those 60 years of age and older. Furthermore, increasing trends of critical care resource use and the incidence of complications were also noted. Therefore, strategies are needed to reduce infections and optimize critical care resource use with greater focus on older people.

## Figures and Tables

**Figure 1 jcm-11-04911-f001:**
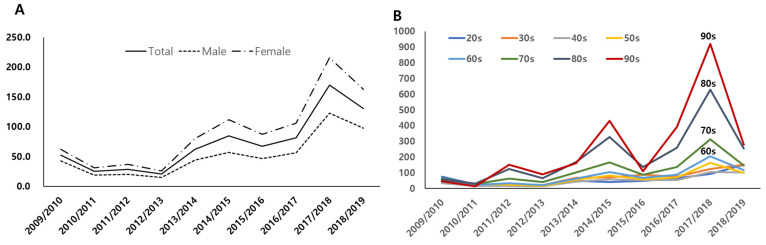
Age-specific annual hospitalization rates (per 100,000 population). (**A**) Annual trend of influenza-related hospitalization rates (*X* axis = years; *Y* axis = number of patients per 100,000 population). (**B**) Annual hospitalization rates by age groups (*X* axis = years; *Y* axis = number of patients per 100,000 population by age groups).

**Figure 2 jcm-11-04911-f002:**
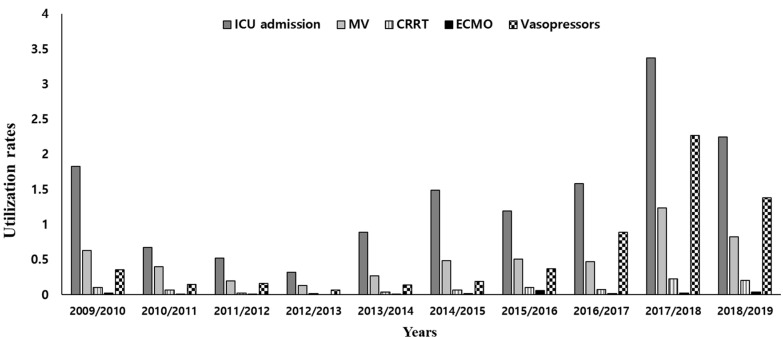
Rates of critical care resource utilization (per 100,000 population). MV: mechanical ventilation; CRRT: continuous renal replacement therapy; ECMO: extracorporeal membrane oxygenation.

**Figure 3 jcm-11-04911-f003:**
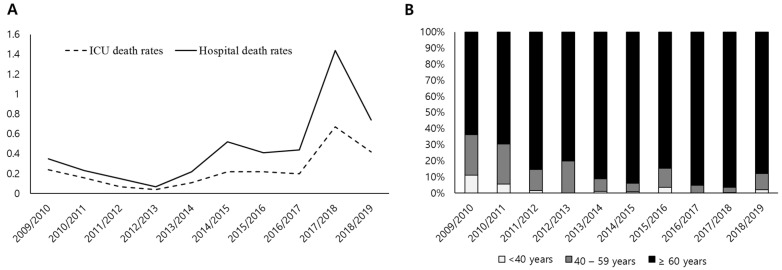
ICU and in-hospital mortality rates. (**A**) Annual trends of ICU and hospital mortality rates per 100,000 population (*X* axis = years; *Y* axis = number of deaths per 100,000 population). (**B**) Annual proportions of three age groups who died in hospitals (i.e., in-hospital death; *X* axis = years; *Y* axis = proportions of age groups). ICU: intensive care unit.

**Table 1 jcm-11-04911-t001:** Influenza-related hospitalization rates and patients’ characteristics.

Variables	2009/2010	2010/2011	2011/2012	2012/2013	2013/2014	2014/2015	2015/2016	2016/2017	2017/2018	2018/2019
No. of patients *	20,465	9860	11,400	8247	25,301	34,986	27,951	34,240	71,944	55,758
Proportion of patients (%) ^†^	0.05	0.03	0.03	0.02	0.06	0.08	0.07	0.08	0.17	0.13
Age-adjusted hospitalization rate ^‡^	52.61	25.04	28.61	20.46	62.06	84.89	67.15	81.51	169.86	130.59
Age, years ^#^	43.8	45.9	53.8	51.8	50.4	55.7	49.6	53.3	57.2	48.6
Age groups (n)										
19–29 years	5807	1987	1246	1023	3639	3038	3589	4810	6881	11,231
30–39 years	3800	2168	2267	1754	5325	5381	6597	5669	9126	10,783
40–49 years	3048	1548	1474	1266	3855	5073	4778	4564	8922	8517
50–59 years	3715	2058	1878	1323	4440	6693	4755	5825	13,746	8440
60–69 years	1948	989	1402	902	2855	5098	3533	4722	11,755	7118
70–79 years	1628	811	1816	1212	3163	5229	2791	4454	10,613	5117
80–89 years	473	283	1138	648	1794	3816	1729	3504	9117	3956
≥90 years	46	16	179	119	230	658	179	692	1784	596
Frequency of men (%)	40.24	37.65	34.64	36.19	34.99	33.36	34.15	34.33	35.89	37.14
Underlying comorbidities (%)										
Diabetes mellitus	2.19	2.94	3.61	4.27	3.46	4.82	3.97	4.07	4.56	3.46
Cardiovascular disease	0.57	0.61	1.22	1.32	0.83	1.21	1.04	1.12	1.43	1.13
Cerebrovascular disease	0.92	1.28	1.77	1.50	1.27	1.82	1.48	1.65	1.91	1.40
Chronic lung disease	1.91	3.04	4.68	4.22	3.48	4.40	4.05	3.78	4.01	3.40
Chronic renal disease	0.61	0.81	0.89	0.96	0.70	1.01	0.88	1.00	1.05	0.95
Chronic liver disease	1.63	2.33	2.99	2.89	2.09	3.31	3.31	2.93	2.95	2.02
Cancer	1.74	1.70	1.68	1.90	1.58	1.79	1.38	1.72	1.75	1.56
Charlson comorbidity index ^#^	0.18	0.23	0.3	0.29	0.23	0.31	0.28	0.28	0.3	0.25
Hospital types (%)										
Tertiary hospital	15.58	13.10	10.47	10.76	7.98	7.31	9.92	9.66	8.78	8.33
General hospital	50.35	47.24	47.73	47.54	46.03	48.59	51.32	47.29	47.97	50.25
Hospital	32.44	38.75	40.48	40.48	44.92	43.09	37.74	42.14	41.91	40.20
Others	1.63	0.90	1.32	1.22	1.08	1.01	1.02	0.92	1.34	1.22
Influenaza vaccination (%) ^‡‡^	28.2	31.9	33.5	35.8	34.0	34.8	35.3	37.5	40.0	41.9

* Influenza season begins as early as October and lasts as late as May of the next year, but all hospitalized patients with influenza diagnosis were collected year-round; ^†^ Proportion of hospitalized patients among the total population in South Korea, *p* < 0.001 by Cochran–Armitage trend test; ^‡^ Age-adjusted rates per 100,000 population; ^#^ Means ± standard deviations or medians with interquartile ranges are presented in the Supporting information; ^‡‡^ Data were extracted from the Health Statistics by the Korea Disease Control and Prevention Agency [21].

**Table 2 jcm-11-04911-t002:** Complications and hospital outcomes (rates per 100,000 population).

Variables	n/Rate	2009/2010	2010/2011	2011/2012	2012/2013	2013/2014	2014/2015	2015/2016	2016/2017	2017/2018	2018/2019	*p*-Value
Myocardial infarct	n	125	32	57	37	93	230	121	173	413	256	<0.001
	rate *^,†^	0.32	0.08	0.14	0.09	0.23	0.56	0.29	0.41	0.98	0.60	
Myocarditis	n	17	1	6	10	12	13	3	8	14	20	0.123
	rate *^,†^	0.04	0.00	0.02	0.02	0.03	0.03	0.01	0.02	0.03	0.05	
Cerebral infarct	n	219	117	195	140	294	582	379	599	1299	651	<0.001
	rate *^,†^	0.56	0.30	0.49	0.35	0.72	1.41	0.91	1.43	3.07	1.52	
Cerebral hemorrhage	n	188	76	148	82	228	401	242	339	717	331	<0.001
	rate *^,†^	0.48	0.19	0.37	0.20	0.56	0.97	0.58	0.81	1.69	0.78	
Liver failure	n	377	163	159	92	250	468	304	295	1022	645	<0.001
	rate *^,†^	0.97	0.41	0.40	0.23	0.61	1.14	0.73	0.70	2.41	1.51	
Heart failure	n	378	156	330	198	451	828	690	1059	2967	1619	<0.001
	rate *^,†^	0.97	0.40	0.83	0.49	1.11	2.01	1.66	2.52	7.00	3.79	
ICU death	n	92	62	27	15	45	89	92	83	284	179	<0.001
	rate *^,†^	0.24	0.16	0.07	0.04	0.11	0.22	0.22	0.20	0.67	0.42	
Hospital death	n	135	89	61	30	91	216	169	184	610	314	<0.001
	rate *^,†^	0.35	0.23	0.15	0.07	0.22	0.52	0.41	0.44	1.44	0.74	
Median length of ICU stay ^‡^	days	10.0	9.0	11.0	12.0	10.0	11.0	12.0	9.0	10.0	10.0	0.003
Median length of hospital stay ^‡^	days	5.0	5.0	4.0	4.0	4.0	4.0	4.0	4.0	4.0	4.0	<0.001

ICU: intensive care unit.; * Rates per 100,000 population. ^†^ Compared by Cochran–Armitage trend test; ^‡^ compared by Kruskal–Wallis test. Medians with interquartile ranges are presented in the Supporting information.

**Table 3 jcm-11-04911-t003:** Univariable and multivariable analyses for in-hospital mortality.

Variables	Univariable Analysis		Multivariable Analysis	
	OR	95% CI	OR	95% CI
Age	1.089	1.086–1.093	1.090	1.085–1095
Gender (male)	1.835	1.676–2.008	1.491	1.339–1.660
Years				
2009/2010–2010/2011	reference	reference	reference	reference
2011/2012–2012/2013	0.625	0.489–0.798	0.350	0.265–0.462
2013/2014–2014/2015	0.687	0.578–0.817	0.487	0.399–0.596
2015/2016–2016/2017	0.767	0.648–0.907	0.383	0.312–0.468
2017/2018–2018/2019	0.979	0.845–1.133	0.378	0.315–0.453
Charlson comorbidity index	1.408	1.386–1.430	1.233	1.207–1.259
Type of hospitals				
Hospitals *	reference	reference	reference	reference
General hospitals ^†^	4.949	4.223–5.800	1.944	1.645–2.298
Tertiary hospitals ^‡^	15.993	13.546–18.883	2.365	1.955–2.860
Others	4.656	3.0370–7.1392	2.907	1.865–4.530
Mechanical ventilation	140.110	126.000–155.799	11.873	10.129–13.916
CRRT	284.161	229.745–351.467	7.233	5.398–9.693
Use of vasopressors	125.171	112.938–138.729	11.248	9.689–13.057
Use of steroids	4.568	4.030–5.177	0.900	0.761–1.065

CI: confidence interval; OR: odds ratio; CRRT: continuous renal replacement therapy; * Hospitals with 31–100 beds. ^†^ Hospitals with >100 beds and >7 specialty departments. ^‡^ Hospitals with >20 specialty departments, serving as a teaching hospital.

## Data Availability

The data presented in this study are available on request from the corresponding author.

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
