# Peer review of "Recent Increases in Influenza-Related Hospitalizations, Critical Care Resource Use, and In-Hospital Mortality: A 10-Year Population-Based Study in South Korea"

_jcm, 2022, doi:10.3390/jcm11164911_

Round 1

Reviewer 1 Report

Thank you enabing me to read your manuscript. The topic is relevant. I have some minor issues:

- I think you arbitrarily determined four (instead of three) types of hospitals: 'hospital'(1), 'general hospital'(2), 'tertiary hospital'(3) and 'other'(4)?
In addition, hospitals were divided into three types according to their supporting levels, which was based on bed capacity and range of specialties as stipulated by the Korean Health Law [19]. A ‘hospital’ was defined as a healthcare facility with >30 beds. A ‘general hospital’ is one with both > 100 beds and > 7 specialty departments, where the inclusion of Internal Medicine, Pediatrics, Surgery, Obstetrics and Gynecology, Pathology, Anesthesiology, and Laboratory Medicine is mandatory. A ‘tertiary hospital’ is defined as one with>20 specialty departments and serves as a teaching hospital. When a healthcare facility did not fall into any of the three types (e.g., nursing hospitals, hospitals for oriental medicine, etc.), it was categorized as ‘others’. 

- Table S2 Hear should be Heart

- in 3.4 you state that you have done multivariate analysis with, among others, 'disease severity' as a variable. Please describe your definition of 'disease severity'. 
I think it is wise to discuss your findings: is it the type of hospital that determines the outcome (in hospital mortality)? This would be a worriesome conclusion for Korean hospitals. You must have your thoughts/specific interpretation of that?

Author Response

Thank you for reviewing the manuscript.
We revised the manuscript according to your comments.
Please see the attachment.

Reviewer 2 Report

I would like to thank the authors for incorporating my suggestions. 

Author Response

Thank you for your comments.

Sincerely yours.

Reviewer 3 Report

Thank you for inviting me to review this manuscript. Hong et al. examined the long-term trends in influenza-related hospitalizations, critical care resource use, in-hospital mortality among adults aged ≥18 years in the Republic of Korea. This study has large sample size and has information over a 10-year period. I have made some major and minor comments below to hopefully improve the manuscript.

Major comments:

·       Throughout manuscript:

o   Would it be possible to present 95% CI instead of p-values.

o   For consistency in language, perhaps describe the results in one way instead of alternating between:

§  Incidence rate of hospitalization and hospitalization rate;

§  Incidence rate of in-hospital mortality and in-hospital mortality rate

·       3.1. Study population:

o   "age"  "mean age" (paragraph 1; line 2)

o   I would describe age by age groups rather than overall mean (SD) age.

o   Provide frequency (%) of men (paragraph 1; line 3)

o   “Diabetes” → “Diabetes mellitus” (paragraph 1line 3)

§  Is this Type-1 and/or Type-2?

·       Table 1:

o   If you have information on vaccination status, I would present that here.

o   I would recommend presenting distribution by age groups (e.g., 18-30, 31-40, 41-50, etc.) instead or in addition to mean (SD) age. I think this is particularly important if you identified older adults as a vulnerable risk group.

o   If you have information on month and/or season of influenza-related hospitalization, I would provide that too. Additionally, you could consider including the season and/or month of influenza-related hospitalization in the footnotes.

·       “3.4. ICU and in-hospital mortality rates”  “3.4. Incidence rates of ICU admission and in-hospital mortality”

o   "While mortalities were lowest during the 2012/2013 season, they gradually increased to peak rates of 0.67/100,000 and 1.44/100,000 population, respectively, in the 2017/2018 season (vs. 0.24/100,000 and 0.35/100,000 population, respectively, in the 2009/2010 season; p < 0.001 and p < 0.001, respectively)" (paragraph 1; lines 2-5)

§  To improve readability for viewers, I would recommend re-wording this sentence (and potentially other sentences) as it was a bit challenging to follow. I suspect it is because the results is mentioning multiple incidence rates of multiple outcomes during multiple influenza seasons.

§  I would present one of two ways:

·       1) "xxx season (ICU: #/100,000; in-hospital mortality: #/100,000)"

·       2) "xxx season (#/100,000 and #100,000, respectively)".

§  E.g. "While ICU admissions and mortalities were lowest during the 2012/2013 season (ICU: #/100,000; in-hospital mortality: #/100,000), they gradually increased to peak rates in the 2017/2018 (ICU: #/100,000; in-hospital mortality: #/100,000) and 2009/2010 (ICU: #/100,000; in-hospital mortality: #/100,000) season; p < 0.001 and p < 0.001, respectively.

·       I would either mention the 2009/2010 season first or remove it entirely. Temporally, incidence rates did not increase 2012/2013 to 2009/2010.

o   "Among the hospitalized patients, a multivariable analysis adjusted for age, gender, comorbidity, and disease severity demonstrated that type of hospitals and the 2009 H1N1 pandemic season (i.e., 2009/2010– 2010/2011) were significantly associated with in-hospital mortality (Table 3)."

§  Describe the association(s) (paragraph 1; lines 7-10)

Minor comments:

·       Throughout manuscript:

o   Be consistent with “outcome”, “hospital outcome” or “patient outcome”. I would stick to one throughout the manuscript.

o   Be consistent with use of space or no space between (>, <, etc.)

§  E.g., “…healthcare facility with >30 beds. A ‘general hospital’ is one with both > 100 beds and > 7 specialty departments…”

·       Title:

o   "Recent Increases in Influenza-Related Hospitalizations, Critical Care Resource Use, and Mortality; a 10-Year Population-Based Study in South Korea"

§   "Recent Increases in Influenza-Related Hospitalizations, Critical Care Resource Use, and In-Hospital Mortality: a 10-Year Population-Based Study in South Korea"

·       Affiliations:

o   Affiliation 3: "Korea"  "Republic of Korea"?

·       Abstract:

o   “outcomes”

§  → “hospital outcomes” or “patient outcomes” whichever you decide to consistently use in the manuscript” (Background; line 2).

o   "Long-term trends in influenza-related hospitalizations, critical care resource use, and outcomes since the 2009 H1N1 influenza pandemic season have been rarely studied." (Background; lines 1-2)

§  That is interesting. Is this overall or for a particular study cohort (i.e., adults, infants, children, elderly, etc.)? Perhaps further clarification would be beneficial.

o   "Both hospitalization and in-hospital mortality rates were primarily driven by patients aged ≥ 60 years." (Results; lines 7-8)

§  → "The high incidence rates of both hospitalization and in-hospital mortality was mainly attributable to patients aged ≥ 60 years"; or

o   “Besides, the incidence of heart failure” (Results; line 15)

§  Use an alternative word to “Besides” or simply remove.

·       1. Introduction:

o   "prepare future" (paragraph 2; line 3)

§   "prepare for future"

o   “Particularly, although influenza cases have decreased for several years after the 2009 H1N1 pandemic season, …”

§  Remove "Particularly" (paragraph 2; line 3)

o   "future policy making"  "guiding future policies" (paragraph 2; line 9)

·       2.1. Data collection

o   "medical practices"  "medical procedures" (paragraph 1; lines 8-9)?

o   "In this study, using the ICD-10 codes" (paragraph 1; line 10).

§  Remove “using the ICD-10 codes” here.

o   "who were hospitalized with influenza diagnosis" (paragraph 1; line 11)

§  → "who were hospitalized with influenza"

o   "This study was approved by the Hallym University Institutional Review Board (approval no. 2020-05-018). Due to the retrospective nature of this study, the decision to obtain written informed consent from patients or their legal surrogates was waived.” (paragraph 2)

§  → “This study and a waiver of consent was approved by the Hallym University Institutional Review Board (approval no. 2020-05-018).”?

·       2.3. Variables for critical care resource use:

o   "counted"  "considered"/"classified" (paragraph 1; line 4)

o   "Treatments"  "Procedures" (paragraph 1; line 8)

o   "counted"  "considered" (paragraph 1; line 17)

·       2.4. Variables for outcomes:

o   Remove "The incidence rates of hospitalization, critical care resource use, complications, and in-hospital death were calculated using mid-year population census data.” (paragraph 1; lines 5-6)

§  This is already more appropriately mentioned under "Data analysis"

·       2.5. Data analysis:

o   Removed "In order" (paragraph 1; line 4)

·       Discussion:

o   "as in the U.S" → "such as in the United States (U.S.)." (paragraph 2; line 6)

o   Remove "during week 34" (paragraph 3; lines 5-6)

o   “lasted until 2010” → “until December 2010”? (paragraph 3; line 6)

o   "United States (U.S.)" > U.S. (paragraph 3; line 8)

o   England → United Kingdom (paragraph 4; line 5)

o   For consistency, "105 "100,000" (paragraph 5; lines 4-6)

Author Response

Thank you for your comments.

We revised the manuscript according to your recommendations.

This manuscript is a resubmission of an earlier submission. The following is a list of the peer review reports and author responses from that submission.

Round 1

Reviewer 1 Report

Authors present trends in influenza-related hospitalizations in South Korea - highly significant given the fact that these hospitalizations seem to increase in the more recent years.

Overall, the paper is well written, and the underlying methods sound if outcome definitions don't suffer too much from classification bias. The biggest contributor could arguably be changing rates of influenza testing. 

It would be helpful if authors could include a few lines in the introduction that, for instance, testing for influenza is standard of care, and thus, influenza ICD-10 coded hospitalizations are likely to be correctly classified. Or if testing is not standard of care, that testing rates are not expected to have changed over the duration of the study (e.g., the underestimation of hospitalization is constant over time). Or if it is not clear if testing rates have changed, explicitly mention in the limitations that year-to-year heterogeneity of influenza test rates could partially explain differences in influenza-related hospitalizations.

Minor point: I find some numbers like Male, Comorbidities and Hospital types in Table 1 hard to compare across years because of different denominators. Consider reporting these as percentages. Consider limiting the number of different (measurement) scales being reported in one table.

Reviewer 2 Report

Thank you for this interesting paper. Hospital capacity and use of resources have been very relevant topics during (seasonal) outbreaks of viral respiratory diseases. Therefore, your manuscript is very relevant.

Some considerations and suggestions:

- Epidemics of influenza and its disease burden vary by different regions [2-6]. Hence, nationwide data from countries with different healthcare systems are needed to establish global strategies and prepare future epidemics or pandemics. -> please explain the Korean health care system in the global perspective. In the discussion, which aspects are relevant for South Korea and which are relevant for the whole world?

- you use 2009 H1N1 pandemic season as a starting point. Why? It is not clear why you have chosen this. Is it because a new influenza type emerged? Or because healthcare infrastructure has changed since then? But if so, how?

- in the methods section, 2.1, you explain influenza season (7-8 months a year). It is unclear if you have included only subjects during this influenza season or year round. Please explain.

- in 2.3: are all these variables relevant to answer your research question? If so, can you discuss the quality of the registration of all these variables, e.g. the comorbidity. That same question comes up when I read 2.4 (variables for outcome).

- the number/percentage of men in the study population is low. Please discuss.

- In 18/19 1374/55758 had diabetes (table 1). That is 2.5%. Google learned me that 'In 2018, 13.8% of Korean adults aged ≥30 years had diabetes, and adults aged ≥65 years showed a prevalence rate of 28%'.
Why do hospitalized patients with influenza do not have diabetes? Or, are the data not correct? The same applies for other comorbidity? Prevalence seems very low in your database. Why?

- 3.3 the high incidence of liver failure after admission due to influenza is striking. heart failure is not in your set of variables (why not?) but seems a more common complication. Could a bias in reporting be a component in the liver failure as most common complication?
Or is it in the definition of liver failure?

- In table 2: please add 'median' or 'mean' before 'length of stay'. 

- in your discussion you declare that your results (with increase in hospitalization in 17/18) is consistent with other reports. Why? Has the severity of that flu season an impact on hospitalisation? Is surveillance of flu present in South Korea and would comparison with these surveillance data be of interest? Without going into details, some discussion about the cause of your findings would be relevant.

- is it that disease severity demands more ICU resources (RRT, ECMO)? Or is its availability (e.g. ECMO machines) responsible for the increase in use?

- in 2.3 'Regarding the steroid use, to include patients who received steroids for acute respiratory failures, only doses of ≥ 40 mg methylprednisolone (or its equivalent doses of other steroids) for ≥ 5 days were counted, based on a previous study [18].' You refer to a study from 2020. Was this policy used in South Korea from 2009-2019? If so (or if not), please reconsider this variable.

- when trends are more or less comparable to what happens in the US or the UK or ...., what does your study add to mondial knowledge. What is new or different? Please explain.